# Experimental Study on Variable-Amplitude Fatigue Performance of M60 High-Strength Bolts of Grid Structures with Bolted Spherical Joints

**DOI:** 10.3390/ma15248939

**Published:** 2022-12-14

**Authors:** Zichun Zhou, Shujia Zhang, Honggang Lei, Bin Qiu, Liang Zhang, Guoqing Wang

**Affiliations:** College of Civil Engineering, Taiyuan University of Technology, Taiyuan 030024, China

**Keywords:** fatigue test, grid structure, bolt–sphere joint, high-strength bolt, stress–fatigue life

## Abstract

The high-strength bolts of grid structures with bolted spherical joints under the action of suspension cranes are at risk of severe fatigue failure. Thus, this paper studies the variable-amplitude fatigue performance of M60 high-strength bolts. The test results for eight specimens in four loading modes are obtained using an Amsler fatigue testing machine. The fatigue life is also estimated based on Miner and Corten–Dolan’s theories, and the applicability of Corten–Dolan’s theory is verified. The fracture induced by the variable-amplitude fatigue is microscopically analyzed using scanning electron microscopy (SEM), revealing the mechanism of the variable-amplitude fatigue failure. Our findings provide valuable experimental data supporting the fatigue life estimation of grid structures with bolted spherical joints in service.

## 1. Introduction

Bolted spherical joints are most commonly used in grid structures. As typical and well-known fabricated steel structures, grid structures with bolted spherical joints have been widely used in engineering, especially in industrial construction with suspended cranes (Figure 1). They offer the outstanding advantages of a fully bolted connection, short construction period, low price, and green construction method, and were designed by the MERO Company, Germany, in 1942. However, the fatigue failure of the high-strength bolts of grid structures with bolted spherical joints should receive our complete attention.

To date, many studies have investigated the fatigue of bolts. For example, Nam [1] studied the fatigue life of bolts with different dimensions under axial loading. Maljaars [2] improved the relevant specifications after analyzing a large amount of data from bolt fatigue tests. Majzoobi [3] investigated the impact of the tread pitch on bolt fatigue and concluded that the fatigue life of coarse threads was higher than that of fine threads. Schaumann [4] tested M72 bolts with huge diameters to analyze the impact of the diameter on the fatigue of the bolts. Note that this bolt type is used for ring flange connections in wind turbines. Bartsch [5] compared extensive experimental data and concluded that the fatigue strength decreased with an increase in the diameter of the bolts. In another study, Jawwad [6] concluded that the stress concentration coefficient was inversely proportional to the eccentricity ratio through an experimental study and finite element analysis (FEA). Hobbs [7] also examined the effect of eccentric cyclic loading on the fatigue life of bolts and found that an increase in local average stress caused by the eccentricity ratio did not affect their fatigue life. Marcelo [8] investigated the fatigue properties of high-strength bolts in different metallurgical conditions and reported that the fatigue limit of the high-strength bolts increased after thread rolling with tempering temperature (TRBHT). Ding [9] also conducted many tensile tests on bolts with different screw-in depths and established a set of equations for estimating the tensile strength of bolted spherical joints with different bolt screw-in depths [10,11]. The influence of multi-axial load and fatigue behavior on fracture surfaces was studied, and a method of measuring and evaluating fracture surfaces was proposed. The fracture surface topography of X8CRNIS18-9 austenitic stainless steel specimens was quantitatively studied and the characteristics of the whole fracture method were determined. Benasciutti [12] introduced relevant studies on the evaluation of the fatigue strength and structural integrity of engineering components subjected to variable amplitudes or random loads.

In addition, Yang [13], Qiu [14], and Zhou [15,16] conducted constant-amplitude fatigue tests on M20, M30, M39, and M60 high-strength bolts of grid structures with bolted spherical joints to study their fatigue performance. They obtained the corresponding stress–fatigue life (S–N) curves. A method to calculate constant-amplitude fatigue was established based on the allowable stress amplitude. The fatigue mechanism of the bolts was revealed via a metallographic analysis, considering the effects of different bolt diameters, stress amplitudes, stress ratios, and loading equipment on the fatigue performance of high-strength bolts.

Under random alternative loads, the fatigue of high-strength bolts is classed as variable-amplitude fatigue. Therefore, Qiu [17] carried out variable-amplitude fatigue tests on 20 M30 high-strength bolts and designed 4 loading modes based on the stress amplitude: low–high, high–low, low–high–low, and high–low–high. The S–N curves of the variable-amplitude fatigue of M30 high-strength bolts were determined based on the equivalent stress amplitude using Miner’s linear cumulative damage theory.

The design specifications for steel structures, such as Eurocode 3 [18], ANSI/AISC 360-16 [19], and GB50017-2017 [20], specify the fatigue strength of high-strength bolts. These specifications have similar criteria for assessing the fatigue performance of standard bolts. However, there are no specific standards for the fatigue performance of high-strength bolts of grid structures with bolted spherical joints.

To this end, this study investigates the fatigue performance of M60 high-strength bolts using a variable-amplitude axial cyclic loading test. The material properties of the high-strength bolts are evaluated after the variable-amplitude fatigue test. A fracture analysis is also conducted to devise a method for estimating the fatigue life of grid structures with bolted spherical joints in service.

## 2. M60 High-Strength Bolts and Bolted Sphere

### 2.1. M60 High-Strength Bolts

In this study, we selected M60 high-strength bolts with a nominal length of 196 mm and a performance grade of 9.8S. They were made of 40Cr steel, and their threads were processed via standard rolling with a pitch of 4 mm. The surface of the bolt was treated with an oxidative blackening treatment, and the effective section area was 2485 mm^2^. Figure 2 illustrates the M60 high-strength bolt used herein.

### 2.2. BS300 Bolted Sphere

The bolted sphere was made of C45 steel with a diameter of 300 mm according to “Bolted Spherical Node of Space Grid Structures” [21] (JG/T 10-2016). No cracks existed on the surface or inside of the bolted sphere. Figure 3 depicts the bolted sphere and its structure.

## 3. Material Properties

The fatigue of the high-strength bolts primarily causes the fatigue of the bolted spherical joints. Thus, investigating the fatigue performance of the high-strength bolts can reflect the fatigue performance of the bolted spherical joints. In this study, we first tested the material properties of the M60 high-strength bolts before fatigue testing to provide a basis for the subsequent variable-amplitude fatigue test so as to determine a reasonable and practical loading range.

Before the fatigue test, three M60 high-strength bolts were selected, randomly numbered, and machined into standard specimens according to “Metallic Materials—Tensile Testing—Part 1: Method of Testing at Room Temperature” [22]. An MTS fatigue testing machine applied static tensile stress to specimens to obtain the stress–strain curve delineated in Figure 4. Table 1 also lists the mechanical properties of the M60 high-strength bolt. According to the relevant indicators given in the specification “High-Strength Bolts for Steel Grid Bolt Ball Nodes” (GB/T 16939-2016) [23], all performance properties of the M60 high-strength bolts complied with the standard for high-strength bolts of grade 9.8 in the specification, satisfying the requirement of the fatigue test.

## 4. Loading Equipment and Scheme

### 4.1. Loading Equipment

The loading equipment mainly comprised an Amsler fatigue testing machine, an M60 high-strength bolt bearing, and a loading frame. The loading devices parameters (Figure 5) were as follows: pulse value: 400 cm^3^/min, five-speed; pulse jacks: four types, including 50/100 kN, 100/200 kN, 250/500 kN, and 500/1000 kN; the pendulum dynamometer meter had four gears, 1/10, 1/4, 1/2, and 1 of the load range.

The loading frame was connected with the upper and lower box-shaped steel beams. The upper steel beam was fixed to a 50 t hydraulic jack at the symmetrical positions on both sides of the steel beam, and the middle part of the equipment was connected with the lower steel beam through the bolted spherical joint. A cyclic tensile stress was applied to the specimen group by compression stress produced by two 50 t hydraulic jacks on the lower steel beam. The M60 high-strength bolt bearing was made of Q345B steel with a slab thickness of 40 mm. There was a bolt hole with a diameter of 65 mm in the middle of the slab, which was used to simulate the connection between the sealing plate and the high-strength bolt of grid structures with bolted spherical joints. The bolt bearing could transfer the load to the specimen.

The load was applied to the specimen group through two hydraulic jacks connected to the Amsler fatigue testing machine, and the compression from the jacks was applied to the specimen group through the steel beam. The cyclic fatigue test could be conducted on two high-strength bolts simultaneously. The fatigue testing machine stopped operating and recording the number of cycles upon the fracture of the M60 high-strength bolt. The loading equipment was self-balancing, meaning it could be considered reasonable and safe. As shown in Figure 5, the loading equipment had good lateral stiffness and coaxiality and operated stably for measuring relevant parameters when repeated variable loads were applied to the specimen.

To ensure the accuracy of the test data, the hydraulic jack should be calibrated before the test. Due to the inevitable deviation in the vertical direction or hole position during sample preparation, the application of cyclic loads will affect the swing of the beam from the plane. To minimize the damage to the fatigue test machine jack and the impact of external factors on the accuracy of the test data, it is necessary to keep the sample upright and on the same plane as the load equipment during the fatigue test.

### 4.2. Loading Scheme

Grid structures with bolted spherical joints are mainly subjected to axial loading; therefore, an axial load was applied to the specimen. In this study, we divided the loading into four modes according to the stress amplitude: high–low (H–L), low–high (L–H), low–high–low (L–H–L), and high–low–high (H–L–H); each mode comprised two or more constant loading stages. To be specific, mode H–L indicated that the stress amplitude changed from high to low, while mode L–H represented the stress amplitude changing from low to high. Mode L–H–L implied that the stress amplitude changed from low to high and then to low, while mode L–H–L represented the stress amplitude changing from low to high and then to low. In each mode, the specimen was subjected to loading until it was completely fractured, and the number of cycles (*N_i_*) for each stress amplitude was recorded. The total fatigue life could be calculated using the equation *N_f_* = *N*_1_ + *N*_2_ + … + *N_i_*. The stress histograms of variable-amplitude fatigue are shown in Figure 6. Table 2 presents the variable-amplitude fatigue test results for the M60 high-strength bolts.

## 5. Analysis of Variable-Amplitude Fatigue Test

During the variable-amplitude fatigue test, all eight M60 high-strength bolts failed at the first thread of the bolts screwed into the bolted sphere. Figure 7 illustrates the M60 high-strength bolts that failed under fatigue loading.

The failure of grid structures with suspended cranes is caused by cumulative fatigue damage from variable-amplitude cyclic loading. The cumulative fatigue damage theory is essential and practical for estimating fatigue life. The commonly used cumulative damage theories are the linear cumulative fatigue damage theory and the nonlinear cumulative fatigue damage theory. This paper used these two theories to analyze the test results.

### 5.1. Miner’s Theory on Fatigue Damage Calculation

Miner’s theory [24] is a typical linear cumulative damage theory, indicating that fatigue damage can be accumulated linearly under cyclic loading and that no interaction exists between the stresses. Failure occurs when the fatigue damage to the specimen or structure accumulates to the ultimate value. The theory is widely adopted in engineering for its simple calculation without the impact of the loading sequence. Its equation is given by:(1)$$D_{M}=\sum_{i=1}^{k}\frac{n_{i}}{N_{i}}$$
where *D_M_* is the cumulative damage value, *n_i_* indicates the number of cycles under stress at grade *I*, and *N_i_* represents the fatigue life under the stress level at grade *i*. It is regarded that the structure fails as *D_M_* equals 1.0.

Miner’s theory was used to analyze the results of the M60 high-strength bolts under variable-amplitude fatigue loading. Table 3 lists the fatigue damage to the M60 high-strength bolts.

### 5.2. Corten–Dolan’s Theory on Fatigue Damage Calculation

Miner’s theory is a linear cumulative damage theory and fails to consider the load sequence, the deviation produced by instantaneous accumulated damage, and the interaction between the stresses. As a result, the estimated results differ markedly from the tested data. Therefore, nonlinear cumulative damage theories have been proposed. Among them, a standard theory is Corten–Dolan’s theory [25], which states that the stress level to which the material is subjected determines the number of cracks forming on the surface of the material. Its equation for estimating the fatigue life is expressed as:(2)$$N_{f}=\frac{N_{1}}{\sum \alpha_{i}{\left(\sigma_{i}/\sigma_{\max}\right)}^{d}}$$
where *N*_l_ is the constant-amplitude fatigue life under the maximum stress, *α_i_* indicates the ratio of the number of cycles of stress *σ_i_* to the total number of cycles, *σ*_max_ denotes the maximum and minimum stress, and *d* represents a material constant determined by the two-level fatigue test. Here, *d* is 4.8 for high-strength steel, so this study sets *d* at 4.8 to derive the equation. The equation is given by:(3)$$D_{CD}=\sum_{i=1}^{i}\frac{\alpha_{i}N_{f}}{N_{1}{\left(\sigma_{\max}/\sigma_{i}\right)}^{d}}$$

Corten–Dolan’s model was used to calculate the fatigue damage (*D_CD_*) to M60 high-strength bolts under variable-amplitude fatigue stress. Table 4 presents the cumulative fatigue damage to the M60 high-strength bolts.

Figure 8 shows the dispersion of the amplitude life prediction values for M60 high-strength bolts using Miner’s rule and Corten–Dolan’s rule. The horizontal coordinate represents the fatigue life values of the bolts obtained from the fatigue test, while the vertical coordinate represents the fatigue life values calculated using the two models.

From Table 1 and Table 2 and Figure 8, the following conclusions could be drawn.

Firstly, as far as the dispersion was concerned, the bolt fatigue life predicted using Corten–Dolan’s rule showed a smaller dispersion. Only one set of data was outside the 2 times life error band, and the rest were within the 1.5 times error band. However, the data predicted by Miner’s rule showed great discreteness, with two datapoints exceeding the error of twice the fatigue life. In general, both models could be used to predict the variable-amplitude fatigue life of M60 high-strength bolts.

Secondly, all of the fatigue life curves predicted by Miner’s rule were above the Y = X line, which indicated that Miner’s rule overestimated the fatigue life of the M60 high-strength bolts. On the other hand, the Corten–Dolan rule underestimated the fatigue life of the high-strength bolts. However, the fatigue life value obtained by the Corten–Dolan rule was closer to the central axis, with three datapoints located near the central axis. This indicated that the Corten–Dolan rule was more suitable for predicting the variable fatigue life of M60 high-strength bolts. In addition, there was an anomaly result (M60-35) in the lower right corner of the graph that was well beyond the 2 times error band. The results showed that the fatigue life of the high strength bolt was much higher than the predicted value obtained by the Corten–Dolan rule. This result may have been because the fatigue test force loading was a slow linear process, during which the force on the high-strength bolt would be less than the theoretical value. It might also have been due to the eccentricity caused in the process of replacing the damaged specimen during the test, which led to the unbalanced force loaded on the two bolts so that the actual force of the bolts was less than the preset value.

In conclusion, in terms of the dispersion and prediction accuracy, the values predicted by the Corten–Dolan rule were more consistent with the actual performance fatigue life of the M60 high-strength bolts than those predicted by Miner’s method. 

The fatigue fracture analysis recorded the whole process from the crack initiation to crack expansion and the fracture of the material under cyclic loading. The morphology, color, roughness, and crack propagation path of the fracture are influenced by the stress state, the properties of the testing material, and the environment. Therefore, it is essential to conduct a fatigue fracture analysis to reveal the fatigue damage mechanism.

Figure 9 depicts the fatigue fracture of specimen M60-7. The overview of fracture surface (Figure 9a), the crack initiation zone (Figure 9b), the crack propagation region (Figure 9c), and the transient fracture zone (Figure 9d) can be observed clearly. The notch at the threads and the severe stress concentration generated by surface inclusions create the fatigue source. The fatigue propagation direction is vertical to that of the principal stress. Further, there is a nearly parallel striped pattern on the surface of the propagation zone. Its surface is smooth and delicate, produced by friction on the fracture surface caused by repeated opening and closing of the cracks. The transient fracture zone has a rough and fibrous surface and a dimple caused by the unstable propagation of the fatigue crack propagated to a critical size. The above findings indicate that the final fracture of the bolts is brittle.

## 6. Conclusions

From the variable-amplitude fatigue failure analysis of M60 high-strength bolts, the following conclusions can be drawn:The variable-amplitude fatigue performance of eight M60 high-strength bolts was successfully analyzed in four loading modes using an Amsler fatigue testing machine;The damage values of the variable-amplitude fatigue tests were calculated utilizing Miner and Corten–Dolan’s theories. The results showed that the *D_M_* values calculated using Miner’s theory showed a significant deviation, while the *D_CD_* values calculated by Corten–Dolan’s theory were close to 1.0 and their dispersion markedly declined. Thus, Corten–Dolan’s theory fully demonstrated its applicability to estimating the variable-amplitude fatigue life of M60 high-strength bolts;The fatigue fracture analyzed via scanning electron microscopy (SEM) presented typical fatigue damage characteristics, showing the fatigue source, the propagation zone, and the transient fracture zone, revealing the failure mechanism of high-strength bolts under variable-amplitude fatigue.

The research results of this paper provide an important reference for establishing the fatigue design method of the bolted spherical joints of a grid structure. Under a complicated load spectrum, the Corten–Dolan model life was more accurate in estimating the fatigue. In future studies, it will be necessary to increase the fatigue tests for M60 high-strength bolts under variable-amplitude loading to better predict the fatigue life.

## Figures and Tables

**Figure 1 materials-15-08939-f001:**
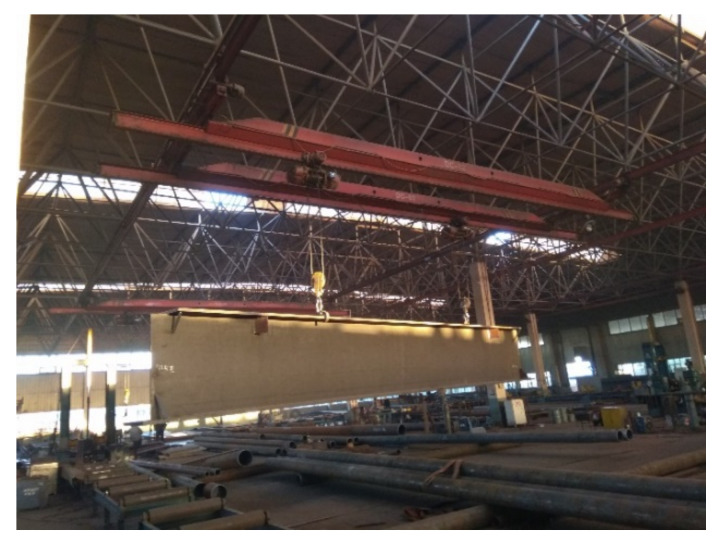
The applications of grid structures with suspended cranes.

**Figure 2 materials-15-08939-f002:**
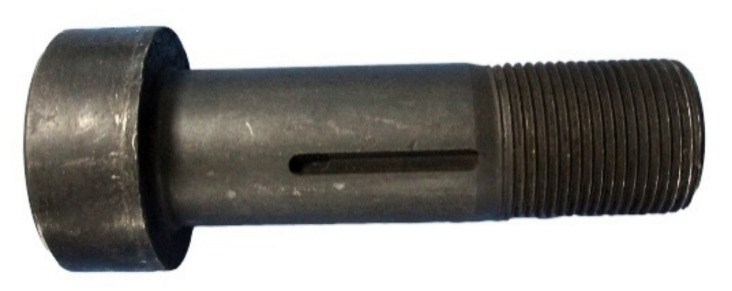
The M60 high-strength bolt used herein.

**Figure 3 materials-15-08939-f003:**
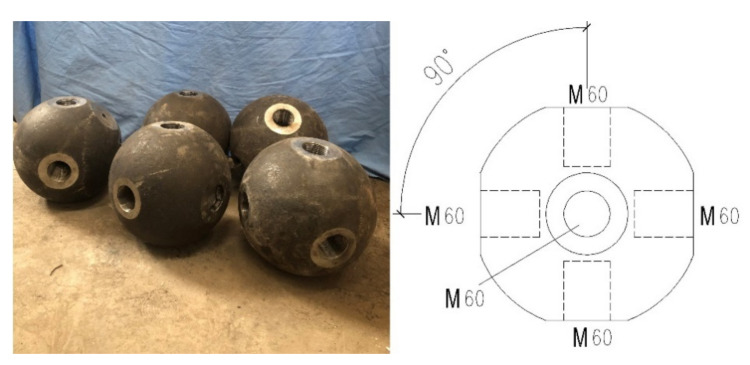
The bolted sphere used herein.

**Figure 4 materials-15-08939-f004:**
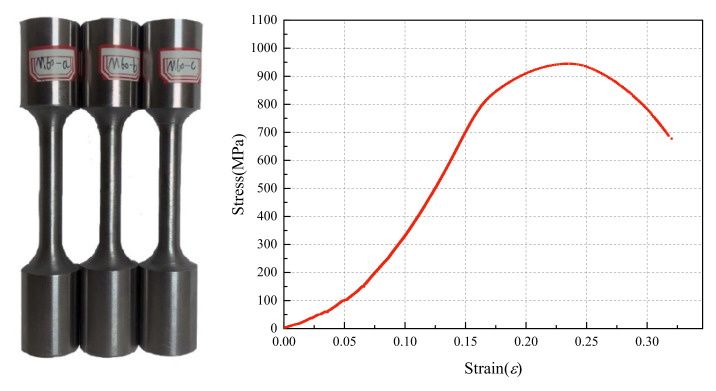
The tensile test on the standard specimens of M60 high-strength bolts.

**Figure 5 materials-15-08939-f005:**
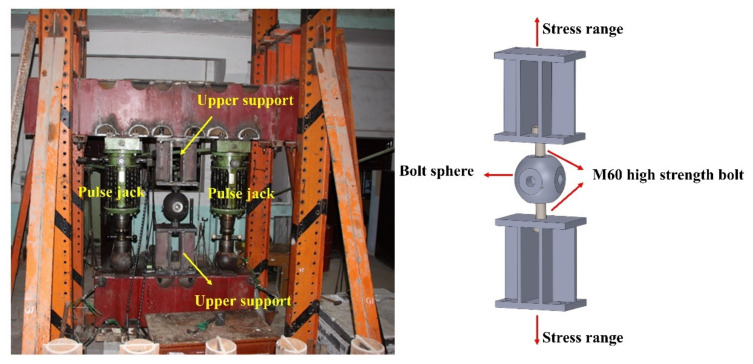
A general view of the fatigue loading of the bolted spherical joint.

**Figure 6 materials-15-08939-f006:**
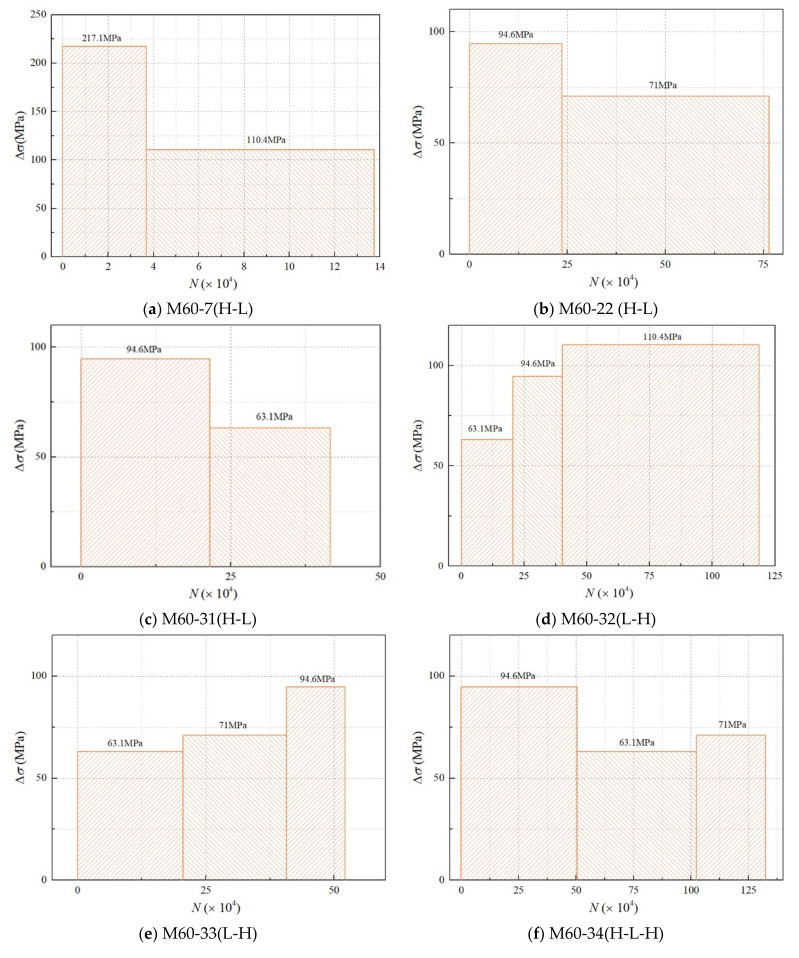
The stress histogram of M60 high-strength bolts under variable-amplitude loading.

**Figure 7 materials-15-08939-f007:**
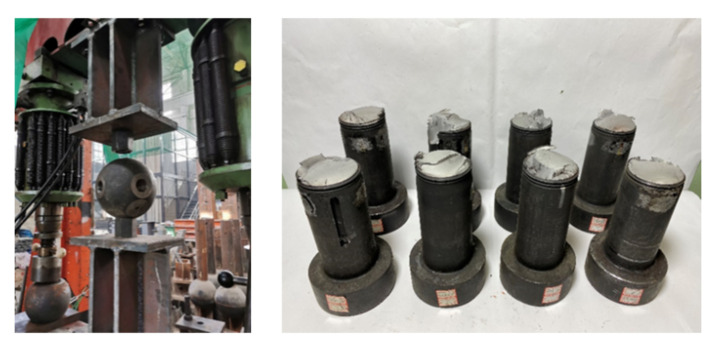
The M60 high-strength bolts that failed under fatigue loading.

**Figure 8 materials-15-08939-f008:**
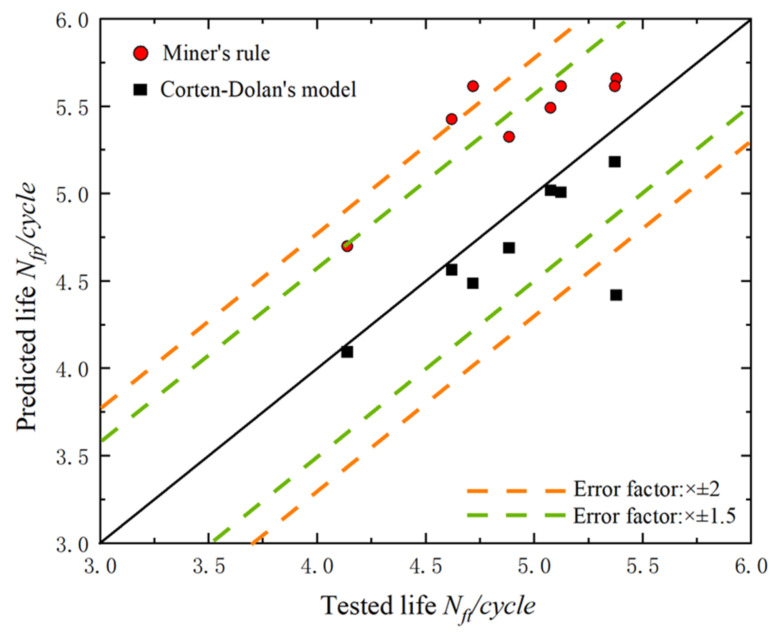
Comparison between fatigue lives predicted using the proposed model, Miner’s rule, and Corten–Dolan’s model, and the fatigue lives tested for M60 high-strength bolts.

**Figure 9 materials-15-08939-f009:**
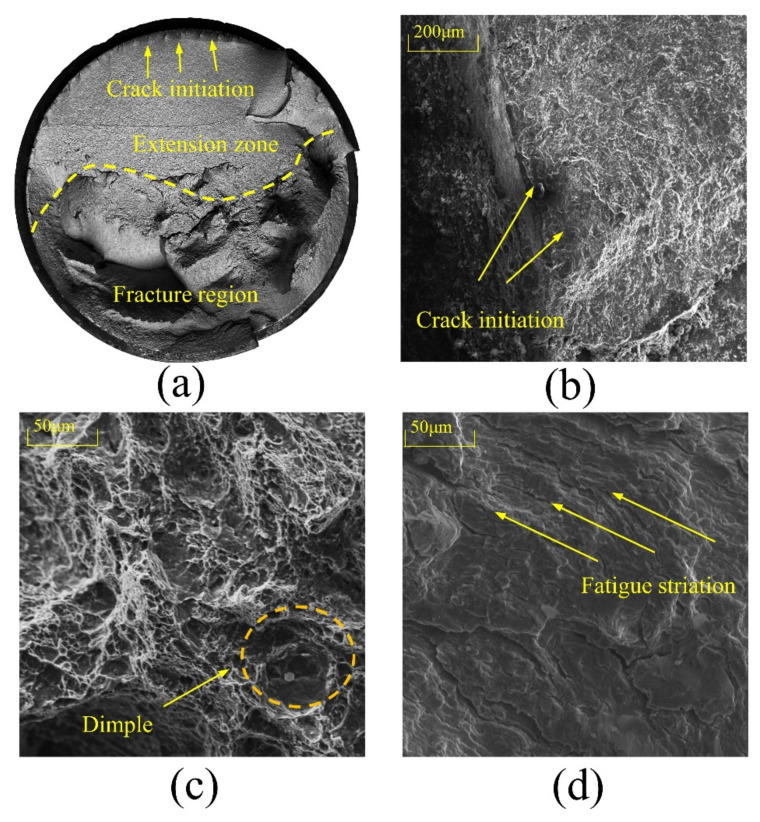
The fatigue fracture of specimen M60-7: (**a**) Overview of fracture surface; (**b**) Crack initiation zone; (**c**) Crack propagation region; (**d**) Transient fracture region.

**Table 1 materials-15-08939-t001:** The mechanical properties of the M60 high-strength bolt material.

Specimen ID	$\sigma_{0.2}$	${\bar{\sigma }}_{0.2}$	$\sigma_{b}$	${\bar{\sigma }}_{b}$	δ(%)	$\bar{\delta }(\%)$
M60-a	930.6	931.0	1047.1	1049.3	17.6	17.2
M60-b	922.3		1068.6		17.8	
M60-c	929.8		1032.2		16.5	

Note: $\sigma_{0.2}$ is the condition yield strength (MPa); ${\bar{\sigma }}_{0.2}$ indicates the average condition yield strength (MPa); $\sigma_{b}$ represents the ultimate tensile strength (MPa); ${\bar{\sigma }}_{b}$ denotes the average ultimate tensile strength (MPa); δ stands for the elongation rate after the fracture; $\bar{\delta }$ is the average elongation rate after the fracture.

**Table 2 materials-15-08939-t002:** The variable-amplitude fatigue test results of the M60 high-strength bolts.

Specimen ID	*σ_max_* (MPa)	*σ_min_* (MPa)	Δ*σ_i_* (MPa)	*N_i_* (×10^4^ Cycles)	*N_f_* (×10^4^ Cycles)	Loading Modes
M60-7	315.7	98.6	217.1	3.70	13.75	H–L
	161.7	51.3	110.4	10.05		
M60-22	138.0	43.4	94.6	23.66	76.47	H–L
	105.5	35.5	71	52.81		
M60-31	138.0	43.4	94.6	21.48	41.63	H–L
	94.6	31.5	63.1	20.15		
M60-32	94.6	31.5	63.1	20.52	118.67	L–H
	138.0	43.4	94.6	19.73		
	161.7	51.3	110.4	78.42		
M60-33	94.6	31.5	63.1	20.55	52.19	L–H
	105.5	35.5	71	20.15		
	138.0	43.4	94.6	11.49		
M60-34	138.0	43.4	94.6	50.46	132.68	H–L–H
	94.6	31.5	63.1	51.88		
	105.5	35.5	71	30.34		
M60-35	94.6	31.5	63.1	96.67	238.67	L–H–L
	138.0	43.4	94.6	114.01		
	161.7	51.3	110.4	7.84		
	105.5	35.5	71.0	20.15		
M60-36	94.6	31.5	63.1	162.64	235.11	L–H–L
	138.0	43.4	94.6	52.37		
	105.5	35.5	71.0	20.10		

**Table 3 materials-15-08939-t003:** The cumulative fatigue damage to the M60 high-strength bolts.

Specimen ID	Δ*σ_i_* (MPa)	*n_i_* (×10^4^ Cycles)	*D_M_*	Loading Mode
M60-7	217.1	3.70	0.777	H–L
	110.4	10.05		
M60-22	94.6	23.66	0.720	H–L
	71.0	52.81		
M60-31	94.6	21.48	0.424	H–L
	63.1	20.15		
M60-32	63.1	20.52	1.910	L–H
	94.6	19.73		
	110.4	78.42		
M60-33	63.1	20.55	2.382	L–H
	71.0	20.15		
	94.6	11.49		
M60-34	94.6	50.46	1.228	H–L–H
	63.1	51.88		
	71.0	30.34		
M60-35	63.1	96.67	2.521	L–H–L
	94.6	114.01		
	110.4	7.84		
	71.0	20.15		
M60-36	63.1	162.64	1.738	L–H–L
	94.6	52.37		
	71.0	20.10		

**Table 4 materials-15-08939-t004:** The cumulative fatigue damage to the M60 high-strength bolts calculated using Corten–Dolan’s theory.

Specimen ID	Δ*σ_i_* (MPa)	*n_i_* (× 10^4^ Cycles)	*D_CD_*	Loading Mode
M60-7	217.1	3.70	1.106	H–L
	110.4	10.05		
M60-22	94.6	23.66	1.563	H–L
	71.0	52.81		
M60-31	94.6	21.48	1.134	H–L
	63.1	20.15		
M60-32	63.1	20.52	1.137	L–H
	94.6	19.73		
	110.4	78.42		
M60-33	63.1	20.55	1.698	L–H
	71.0	20.15		
	94.6	11.49		
M60-34	94.6	50.46	1.300	H–L–H
	63.1	51.88		
	71.0	30.34		
M60-35	63.1	96.67	9.079	L–H–L
	94.6	114.01		
	110.4	7.84		
	71	20.15		
M60-36	63.1	162.64	1.541	L–H–L
	94.6	52.37		
	71.0	20.10		

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
