# Peer review of "Experimental Study on Variable-Amplitude Fatigue Performance of M60 High-Strength Bolts of Grid Structures with Bolted Spherical Joints"

_materials, 2022, doi:10.3390/ma15248939_

Round 1

Reviewer 1 Report

The manuscript “Experimental Study on Variable-Amplitude Fatigue Performance of M60 High-Strength Bolts of Grid Structures with Bolted Spherical Joints” studied the variable-amplitude fatigue performance of M60 high-strength bolts. The test results of eight specimens in four loading modes were obtained using an Amsler fatigue testing machine. Although the topic addressed is relevant, the manuscript as presented cannot be accepted for publication. This must undergo a wide review. Among the biggest concerns are the lack of clarity in the scientific contribution and the result superficial analysis. One can also mention the lack of care with the figures and tables, which are of a quality far below that necessary for a manuscript to be published in Materials. Below are some comments.

1. The caption for figure 1 must be on the same page as the figure 1.

2. ANSI/AISC 360-16, GB50017-2017 and GB/T 16939-2016 should be added to the reference list.

3. The quality of figures 4, 5 and 7 should be improved.

4. Have the mechanical properties of the M60 high strength screws shown in Table 1 been measured?

5. Is the Amsler1200 pulse fatigue testing machine calibrated? How to ensure that test results are valid?

6. What is the resolution of the Amsler1200 pulse fatigue testing machine? The declaration of the measurement system resolution is essential since the number of significant digits in the measurement result depends on resolution.

7. Check the number of significant digits in the results presented in tables 2, 3 and 4. The table 2 must be on one page. The same comment applies to Tables 3 and 4.

8. Use the word "equation" in place of "formula". Cite the equations in the text before presenting them.

9. On page 8 “Both are close to 1.0, and the deviation is significantly reduced,”

a) What does the word deviation mean? Comment!

10. No statistical analysis such as ANOVA or a non-parametric test was carried out to analyze and conclude whether the four loading modes evaluated influence the variable-amplitude fatigue performance of M60 high-strength bolts.

The results must be analyzed comprehensively. These should be compared with results already published by other authors.

11. It was not clear what the scientific contribution of the manuscript wa

Reviewer 2 Report

Comments and Suggestions for Authors

The paper reports an interesting and very useful experimental work covering fatigue failure behavior of grid structures with bolted spherical joints. The manuscript is well structured and can be published after some revisions. The reviewer enjoyed reading this paper.

The manuscript has some weaknesses. Mentioned below aspects should be taken into consideration during the revision:

1.    Units and abbreviations:

I suggest adding "Nomenclature" (as list of symbols, list of abbreviations and subscripts and others) in the manuscript.

2.    Introduction:

Literature analysis should be expanded. It is recommended to better justify variable amplitude loading tests. See for example the fallowing papers:

- https://doi.org/10.1016/j.measurement.2021.109443 - a fractographic study exploring the fracture surface topography of S355J2 steel after pseudo-random bending-torsion fatigue tests;

- https://doi.org/10.3390/met12060919 - fracture, fatigue, and structural integrity of metallic materials and components undergoing random or variable amplitude loadings.

3.    Experimental program:

- Do the authors have more fracture surface topography measurements results? Characteristic zones will surely show differences for parameters defined in accordance with ISO 25178 standard.

- Please plot the sample load signal. This will visualize how the load is applied to the reader.

4.    Conclusions:

- Please, can you change the title of this section to "Concluding Remarks"?

- The conclusions should be in a quantified form.

- The practical usefulness of the results should be emphasized.

- The main limitations of the present method must be identified and discussed in the end of this section.

Round 2

Reviewer 1 Report

Authors addressed appropriately my comments. I recommend the publication of the manuscript. Based on the author response, two additional minor comments are presented.

On Page 5, Line 158 “Pulse value: 400 cm3/min, five-speed; Pulse jack: four types including 50/100 kN, 100/200 kN, 250/500 kN, and 500/1000 kN; The pendulum dynamometer meter has four gears, 1/10, 1/4, 1/2, and 1.”

a) Check the writing of 400 cm3/min.

b) Do pendulum dynamometer meter gears (1/10, 1/4, 1/2, and 1) values have units? Check!

Author Response

Point 1: On page 5, line 158 "Pulse value: 400 cm3/min, five speeds; pulse jacks: 50/100 kN, 100/200 kN, 250/500 kN, 500/1000 kN four types; pendulum dynamometer The machine meter has four gears, 1/10, 1/4, 1/2 and 1.

a) Check the writing of 400 cm3/min.

b) Do pendulum dynamometer gear (1/10, 1/4, 1/2 and 1) values ​​have units? an examination!

Response 1: Thank you for pointing out these issues in your manuscript.

a) We have corrected it to "400 cm3/min"

b) We corrected it to "..., 1/10, 1/4, 1/2 and 1 of the load range".

Reviewer 2 Report

The authors attempted to provide a revision manuscript according to the reviewers' comments. They also responded to all cases individually. Although not all of their answers were satisfactory, they are generally acceptable in the scoring.

Author Response

Thank you for your careful reading, helpful comments, and constructive suggestions, which greatly improved the way our manuscript is presented. We will improve the content in future work.